# Dual Representation Learning
# for Out-of-distribution Detection

**Zhilin Zhao**                                                          *zhaozhl7@hotmail.com*
*Data Science Lab, Macquarie University, Sydney, NSW 2109, Australia*

**Longbing Cao**                                                        *longbing.cao@mq.edu.au*
*Data Science Lab, Macquarie University, Sydney, NSW 2109, Australia*

**Reviewed on OpenReview:** *https://openreview.net/forum?id=PHAr3q49h6*

## Abstract

To classify in-distribution samples, deep neural networks explore strongly label-related information and discard weakly label-related information according to the information bottleneck. Out-of-distribution samples drawn from distributions differing from that of in-distribution samples could be assigned with unexpected high-confidence predictions because they could obtain minimum strongly label-related information. To distinguish in- and out-of-distribution samples, Dual Representation Learning (DRL) makes out-of-distribution samples harder to have high-confidence predictions by exploring both strongly and weakly label-related information from in-distribution samples. For a pretrained network exploring strongly label-related information to learn label-discriminative representations, DRL trains its auxiliary network exploring the remaining weakly label-related information to learn distribution-discriminative representations. Specifically, for a label-discriminative representation, DRL constructs its complementary distribution-discriminative representation by integrating diverse representations less similar to the label-discriminative representation. Accordingly, DRL combines label- and distribution-discriminative representations to detect out-of-distribution samples. Experiments show that DRL outperforms the state-of-the-art methods for out-of-distribution detection.

## 1 Introduction

Deep neural networks demonstrate a significant classification generalization ability Krizhevsky et al. (2012) on samples drawn from an unknown distribution (i.e., *in-distribution*, ID), benefiting from their powerful ability to learn representations Bengio et al. (2013); Gong et al. (2022). From the information-theoretic viewpoint Saxe et al. (2018), for an ID sample, a pretrained network effectively learns the *label-discriminative representation* by exploring strongly label-related information and discarding weakly label-related information. However, test samples could be drawn from distributions different from that of ID samples Ren et al. (2019); Haas et al. (2023) (i.e., *out-of-distribution*, OOD), and the pretrained network is not sensitive to such OOD samples and make unexpected high-confidence predictions on them. It causes that the pretrained network cannot distinguish ID and OOD samples. This over-confidence phenomenon on OOD samples makes networks vulnerable to attack Goodfellow et al. (2015) and causes distributional vulnerability Zhao et al. (2022).

We argue some fundamental causes of the above problem in a pretrained network include: (1) the label-discriminative representations from strongly label-related information focus on capturing the ID input-label mapping, while (2) the network overlooks the learning of *distribution-discriminative representations* from the weakly label-related information that can distinguish ID and OOD. This argument can be explained by the information bottleneck principle Alemi et al. (2017); Wang et al. (2022) of learning a tradeoff between input compression and its label prediction. As a result, the pretrained network only or mainly focuses

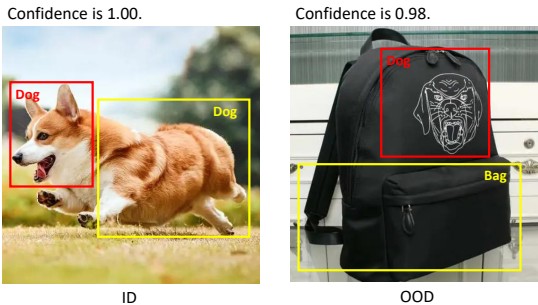

Figure 1: Different informativeness properties of ID and OOD samples. Red and yellow boxes capture the information for learning label- and distribution-discriminative representation, respectively. The label- and distribution-discriminative representations of the ID sample correspond to the same label (dog). Conversely, the label- and distribution-discriminative representations of the OOD sample correspond to different labels (dog and bag). However, the OOD sample is assigned with a high-confidence prediction for the wrong label (dog) since it obtains minimum strongly label-related information about the label. It causes networks cannot distinguish between ID and OOD samples.

on strongly label-related information but discards weakly label-related information. Strongly and weakly label-related information are complementary, and their corresponding label- and distribution-discriminative representations contain labeling information for an ID sample. An ID sample has both strongly and weakly label-related information, and an OOD sample with minimum strongly label-related information still receives a high-confidence prediction from the pretrained network. For example, if a pretrained network merely focuses on head (strongly label-related information) and ignores body (weakly label-related information) to recognize dog images, an OOD sample with minimum information about 'head' could have a high-confidence prediction about the dog class, as shown in Figure 1.

Accordingly, both the label- and distribution-discriminative representations of an ID sample correspond to the same label, and the distribution-discriminative representation of an OOD corresponds to another label or even none of any labels. Therefore, the label-discriminative representations are sufficient for the classification task but insufficient for OOD detection, and the labeling consistency between label- and distribution-discriminative representations could distinguish ID and OOD samples. This inspiration from the different informativeness properties of ID and OOD samples can be summarized as:

> *By dually learning label- and distribution-discriminative representations from strongly and weakly label-related information, respectively, a new perspective is to couple both representations to detect OOD samples.*

Several efforts have been made in the literature to improve the OOD sensitivity of a pretrained network when training OOD samples are unavailable Hendrycks & Gimpel (2017); Sastry & Oore (2020); Lee et al. (2018a); Shalev et al. (2018), including post-hoc and confidence enhancement methods. In *post-hoc methods*, a pretrained network is solely trained on ID samples without considering the predictions for OOD samples, an OOD detector is then trained for the pretrained network. The OOD detector does not acquire new knowledge from training ID samples, making the OOD detection performance heavily dependent on the knowledge about label-discriminative representations learned by the pretrained network Hendrycks et al. (2019a). To address this problem, the *confidence enhancement methods* improve the OOD sensitivity of a pretrained network by retraining or finetuning it with prior knowledge about OOD samples Lee et al. (2018a); Hendrycks et al. (2019b); Tack et al. (2020); Hsu et al. (2020). They regulate the output distribution of the network to encourage high- and low-confidence predictions on ID and OOD samples, respectively Malinin & Gales (2018). This approach enforces the network to extract more information about ID samples through retraining, while it relies heavily on prior knowledge about OOD samples and may not be applicable to unknown OOD samples. As a result, the existing post-hoc and confidence enhancement methods suffer from limited OOD detection performance. To address the limitations of these two methods, we learn OOD-sensitive representations (i.e.,

distribution-discriminative representations) from the weakly label-related information of ID samples and distinguish ID and OOD samples according to their different informativeness properties.

By exploring the different informativeness properties of ID and OOD samples, we propose a *dual representation learning* (DRL) approach to learn both label- and distribution-discriminative representations from strongly and weakly label-related information, respectively. For the generality and flexibility of DRL, we assume a pretrained network is given which only focuses on learning the label-discriminative representations to classify ID samples. We introduce an *auxiliary network* to learn the complementary distribution-discriminative representations corresponding to the label-discriminative representations. Therefore, the pretrained network is trained solely on ID samples, while the auxiliary network is trained on both ID samples and the corresponding label-discriminative representations from the pretrained one. The pretrained and auxiliary networks aim to extract label-related information from inputs to learn representations. Specifically, the label- and distribution-discriminative representations learned from the two networks are strongly and weakly related to labeling, respectively. To ensure the above properties of the two representations, given an ID sample, we set a constraint so that its distribution-discriminative representation is significantly different from its label-discriminative representation and sensitive to the same label. An implicit constraint is incorporated into the auxiliary network, which integrates multiple representations different from a label-discriminative representation into a complementary distribution-discriminative representation. After learning the auxiliary network depending on the pretrained network, DRL then couples the two representations to calculate an OOD score for each test sample. Accordingly, the OOD scores are then used to distinguish ID and OOD samples for OOD detection.

The main contributions of this work include:

- Taking the information bottleneck principle, we reveal that learning a label-discriminative representation by a pretrained network alone may not sufficiently capture all labeling information of an ID sample. There may exist a complementary distribution-discriminative representation capturing the remaining labeling information.

- We infer the information bottleneck principle to learn the complementary distribution-discriminative representations. Accordingly, we train an auxiliary network to learn a distribution-discriminative representation by integrating multiple representations different from a label-discriminative representation.

- By exploring the different informativeness properties of ID and OOD samples, the label and distribution-discriminative representations are coupled to form OOD scores for distinguishing ID and OOD samples.

The paper is organized as follows. Section 2 reviews the related work. Section 3 presents the proposed DRL method. The experimental results are shown in Section 4, and Section 5 makes some concluding remarks.

## 2 Related Work

The test samples for a DNN could be ID or OOD, and the task of OOD detection Ren et al. (2019) is to differentiate ID and OOD samples. Unlike outlier detection Zong et al. (2018); Schmier et al. (2023) which filters out training samples deviating from the majority through training a detector, OOD detection detects samples different from training ID samples in the test phase for trained networks. In contrast to the open-set recognition Zhang et al. (2020) whose training dataset contains samples from known and unknown classes and which refuses samples from unknown classes in the test phase, OOD detection only accesses a training ID dataset with the corresponding pretrained network learned from this dataset. For a pretrained network, the existing methods of improving the OOD sensitivity include post-hoc and confidence enhancement methods. Post-hoc methods train an OOD detector measuring data relationships sensitive to OOD samples according to the network outputs from a pretrained network, while confidence enhancement methods retrain a pretrained network to improve the OOD sensitivity with prior knowledge about OOD samples.

## 2.1 Post-hoc Methods

A baseline method in Hendrycks & Gimpel (2017) treats the confidence represented by maximum softmax outputs as the OOD score for a test sample, where ID and OOD samples are expected to own high and low scores, respectively. An out-of-DIstribution detector for Neural networks (ODIN) Liang et al. (2018) enhances the baseline method by considering the adversarial perturbation on inputs Miyato et al. (2019) and temperature scaling on the softmax function, and the basic assumption is that the two operations can enlarge the confidence between the ID and OOD samples. Energy-based Detector Liu et al. (2020) (ED) applies the negative energy function in terms of the denominator of the softmax activation for OOD detection, and the log of the confidence in the baseline method is a particular case of the negative energy function. Apart from utilizing the network outputs from the last layer, Mahalanobis Distance-based Detector Lee et al. (2018b) (MDD) adds a small noise perturbation to input and combines its Mahalanobis distances of latent features from different network layers as the OOD score. Rectified Activations (RA) Sun et al. (2021) truncates activations on the penultimate layer of a network to reduce the negative effect of noise. Based on MDD, Gaussian mixture based Energy Measurement Morteza & Li (2022) is a scoring function that is proportional to the log-likelihood of the ID samples and extended to deep neural networks by applying the features extracted from the penultimate layers to calculate scores. However, these detectors are merely trained from the outputs of the network without learning any new knowledge from training ID samples, which indicates that the OOD detection performance heavily depends on the knowledge learned by the pretrained network.

## 2.2 Confidence Enhancement Methods

To explore more information from ID samples to enhance the OOD sensitivity of a pretrained network, another approach is to retrain the pretrained network with prior knowledge about OOD samples. One special idea is to generate OOD samples from training ID samples and apply them to restrict network behaviors on OOD samples. For example, the Joint Confidence Loss (JCL) Lee et al. (2018a) utilizes a generative adversarial network Goodfellow et al. (2014) to generate samples around the boundary of training ID samples and encourage their label probabilities to satisfy a uniform distribution. Instead of generating OOD samples, Self-Supervised Learning (SSL) Hendrycks et al. (2019b) augments an ID sample by rotating it $0°, 90°, 180°, 270°$, respectively, and learns the rotation angles and the labels of augmented ID samples simultaneously. Applying the same augmentation method, the Contrasting Shifted Instances (CSI) Tack et al. (2020) treats the original and augmented ID samples as positive and negative samples in a contrastive loss, respectively. Although these confidence enhancement methods introduce extra knowledge to encourage networks to learn more information sensitive to OOD samples from ID samples, the prior knowledge about OOD samples could be misleading due to the complexity of unknown test samples. From a model perspective, DeConf-C Hsu et al. (2020) believes the softmax function causes a spiky distribution over classes. It addresses this problem by using a divisor structure that decomposes confidence. DeConf-C also applies augmented ID samples that are perturbed by gradient directions. However, it is hard to select an appropriate specific divisor structure for a training ID dataset due to the lack of knowledge about data characteristics. Based on the experimental observation that the OOD detection performance declines as the number of ID classes increases, MOS Huang & Li (2021) groups training ID samples according to label concepts. However, the taxonomy of the label space is usually unavailable, and applying K-Means clustering on feature representations and random grouping to divide the ID dataset cannot ensure that the samples in a group have similar concepts. Based on the assumption that OOD samples are relatively far away from the ID samples, KNN+ Sun et al. (2022) computes the $k$-th nearest neighbor distance between the embedding of each test image and the training set as the OOD score. Compactness and Dispersion Regularized learning Ming et al. (2023) makes a trade off between promoting large angular distances among different class prototypes and encouraging samples to be close to their class prototypes.

# 3 Dual Representation Learning

In this section, we present the proposed DRL method. We assume each ID sample $(\mathbf{x}, y)$ follows the joint distribution $P(\mathcal{X}, \mathcal{Y})$ where $\mathbf{x}$ is an input and $\mathbf{y}$ is its corresponding ground truth label. The number of classes is $K$, and we thus have $y \in [K]$. We assume a pretrained network $g_\phi$ with the parameter $\phi$ transforms an

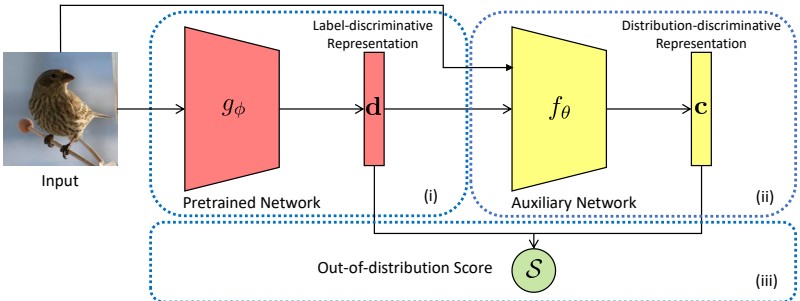

Figure 2: The DRL learning process. It includes: (i) learning a pretrained network on training ID samples for label-discriminative representations; (ii) learning an auxiliary network on training ID samples and their corresponding label-discriminative representations for distribution-discriminative representations; and (iii) calculating the OOD scores by combining these two representations.

input $\mathbf{x} \in \mathcal{X}$ into a label-discriminative representation $g_\phi(\mathbf{x}) = \mathbf{d} \in \mathcal{D}$. We assume an auxiliary network $f_\theta$ with the parameter $\theta$ transforms this sample with its label-discriminative representation $\mathbf{d}$ into the complementary distribution-discriminative representation $f_\theta(\mathbf{x}, \mathbf{d}) = \mathbf{c} \in \mathcal{C}$. Both the pretrained and auxiliary networks own the same network structure. For each input $\mathbf{x}$, we combine the two representations to calculate an OOD score $\mathcal{S}(\mathbf{x})$. Different from traditional ensemble methods Guyon et al. (2017) that combine the outputs from different independent networks, DRL integrates two complementary representations from two dependent networks. Specifically, the auxiliary network learning complementary distribution-discriminative representations depends on the pretrained network learning label-discriminative representations, as shown by the learning process in Figure 2.

Recall that a pretrained network merely learns label-discriminative representations from strongly label-related information, which causes that this pretrained network assigns unexpected high-confidence predictions on OOD samples. Therefore, DRL applies an auxiliary network to learn complementary distribution-discriminative representations from weakly label-related information to improve the OOD sensitivity of the pretrained network. In the following, we firstly reveal the existence of complementary information which is weakly related to labeling. We then describe the information bottleneck principle of extracting weakly label-related information. Based on the information bottleneck principle, we propose the learning method of the auxiliary network. Accordingly, we combine both label- and distribution-discriminative representations to calculate the OOD scores for test samples.

### 3.1 Learning Principle of Label-Discriminative Representations

The information bottleneck principle Alemi et al. (2017) makes a tradeoff between the compression of input and the prediction of its label, and the mutual information Belghazi et al. (2018); Gonzalez-Lopez et al. (2020) measures the shared information between variables. Accordingly, for a network, the process of extracting label-related information from inputs to learn the corresponding representations can be interpreted from the information-theoretic view Saxe et al. (2018).

Given a dataset $\mathcal{X}$ and its label set $\mathcal{Y}$, an ID sample $\mathbf{x}$ contains all the information about its corresponding label $\mathbf{y}$. The information bottleneck limits the information to predict $\mathbf{y}$ by compressing $\mathbf{x}$ to learn its label-discriminative representation $\mathbf{d}$. Therefore, the information of the label-discriminative representations $\mathcal{D}$ shares with the labels $\mathcal{Y}$ should be maximized while the information between $\mathcal{D}$ and the inputs $\mathcal{X}$ should be minimized, i.e.,

$$\max \mathcal{I}(\mathcal{D}; \mathcal{Y}) - \beta_\mathcal{D} \mathcal{I}(\mathcal{X}; \mathcal{D}), \tag{1}$$

where $\mathcal{I}(\cdot; \cdot)$ refers to the mutual information, and $\beta_\mathcal{D}$ controls the trade-off between learning more information from the labels $\mathcal{Y}$ and retaining less information of the original inputs $\mathcal{X}$ for learning label-discriminative

representations $\mathcal{D}$. Specifically, $\mathcal{I}(\mathcal{D}; \mathcal{Y})$ determines how much label information is accessible from the label-discriminative representation, and $\mathcal{I}(\mathcal{X}; \mathcal{D})$ denotes how much information the label-discriminative representation can acquire from the original input.

Note that $\mathbf{x}$ contains all the information of $\mathbf{d}$ because $\mathbf{d}$ is a representation learned from $\mathbf{x}$. Therefore, we have

$$\mathcal{I}(\mathcal{X}; \mathcal{Y}) = \mathcal{I}(\mathcal{X}, \mathcal{D}; \mathcal{Y}), \tag{2}$$

where $\mathcal{I}(\mathcal{X}, \mathcal{D}; \mathcal{Y})$ denotes the shared information between the labels $\mathcal{Y}$ and the union of $\mathcal{X}$ and $\mathcal{D}$. According to the chain rule Federici et al. (2020) of the mutual information, we have,

$$\mathcal{I}(\mathcal{X}; \mathcal{Y}) = \mathcal{I}(\mathcal{D}; \mathcal{Y}) + \mathcal{I}(\mathcal{X}; \mathcal{Y}|\mathcal{D}), \tag{3}$$

where $\mathcal{I}(\mathcal{X}; \mathcal{Y}|\mathcal{D})$, representing the shared information between $\mathcal{X}$ and $\mathcal{Y}$ given the label-discriminative representations $\mathcal{D}$, is greater than or equals 0. Accordingly, we have $\mathcal{I}(\mathcal{X}; \mathcal{Y}) \geq \mathcal{I}(\mathcal{D}; \mathcal{Y})$. Therefore, for an ID input $\mathbf{x}$, its label-discriminative representation $\mathbf{d}$ is insufficient for $\mathbf{y}$ because $\mathbf{d}$ does obtain all the labeling information about $\mathbf{y}$.

Accordingly, we assume there exists another representation for an ID sample $\mathbf{x}$ different from the label-discriminative representation and containing the remaining information about $\mathbf{y}$, i.e., a complementary distribution-discriminative representation $\mathbf{c}$. The distribution-discriminative representations $C$ also satisfies Eq. (2) and Eq. (3) since $\mathbf{c}$ is also a representation of the input $\mathbf{x}$. We can thus obtain the following equation by simply replacing $\mathcal{D}$ in Eq. (3) with $\mathcal{C}$,

$$\mathcal{I}(\mathcal{X}; \mathcal{Y}) = \mathcal{I}(\mathcal{C}; \mathcal{Y}) + \mathcal{I}(\mathcal{X}; \mathcal{Y}|\mathcal{C}). \tag{4}$$

Since $\mathbf{d}$ is insufficient for selecting $\mathbf{y}$ and these is no shared information between $\mathbf{c}$ and $\mathbf{d}$, we thus assume $\mathcal{I}(\mathcal{X}; \mathcal{Y}|\mathcal{D}) = \mathcal{I}(\mathcal{C}; \mathcal{Y})$ and have the following equation according to Eq. (3) and Eq. (4),

$$\mathcal{I}(\mathcal{X}; \mathcal{Y}) = \mathcal{I}(\mathcal{D}; \mathcal{Y}) + \mathcal{I}(\mathcal{C}; \mathcal{Y}). \tag{5}$$

According to Eq. (5), both the label- and distribution-discriminative representations about the same label co-exist for an ID sample, and a label-discriminative representation alone cannot contain all the label information of an ID sample. Specifically, for ID samples, its label-discriminative representation is strongly related to labeling, while its distribution-discriminative representation learned from the remaining label information is weakly related to labeling. Recall that an OOD sample with high-confidence prediction owns a label-discriminative representation that is sensitive to a label and a distribution-discriminative representation that corresponds to other labels or even none of any labels. Accordingly, we can distinguish ID and OOD samples according to different informativeness properties by exploiting the label-discriminative representations and exploring the distribution-discriminative representations.

## 3.2 Learning Principle of Distribution-Discriminative Representations

We present the information bottleneck principle of learning distribution-discriminative representations. Recall that, for an ID sample $\mathbf{x}$, the pretrained network $g_\phi$ learns its label-discriminative representation $\mathbf{d}$ to predict the label $\mathbf{y}$ based on the information bottleneck principle in Eq. (1). Furthermore, the corresponding distribution-discriminative representation $\mathbf{c}$ also contains the information about the label $\mathbf{y}$ because $\mathbf{d}$ is insufficient to contain all label information, which indicates that $\mathcal{C}$ also follows the learning principle of Eq. (1). Further, according to Eq. (5), there is no shared label information between the two kinds of representations, i.e., the mutual information $I(\mathcal{D}, \mathcal{C})$ is expected to be equal to zero. However, in practice, it is impossible to separate the information of the two representations entirely. Based on Eq. (1), we thus minimize the amount of shared information $I(\mathcal{D}, \mathcal{C})$ and have

$$\max \mathcal{I}(\mathcal{C}; \mathcal{Y}) - \beta_\mathcal{C} \mathcal{I}(\mathcal{X}; \mathcal{C}) - \alpha \mathcal{I}(\mathcal{D}; \mathcal{C}). \tag{6}$$

Similar to $\beta_\mathcal{D}$, $\beta_\mathcal{C}$ controls the amount of information propagated from $\mathcal{X}$ to $\mathcal{C}$. A larger $\beta_\mathcal{C}$ will lead to more information being extracted from $\mathcal{X}$, which also indicates more label-unrelated information will be extracted

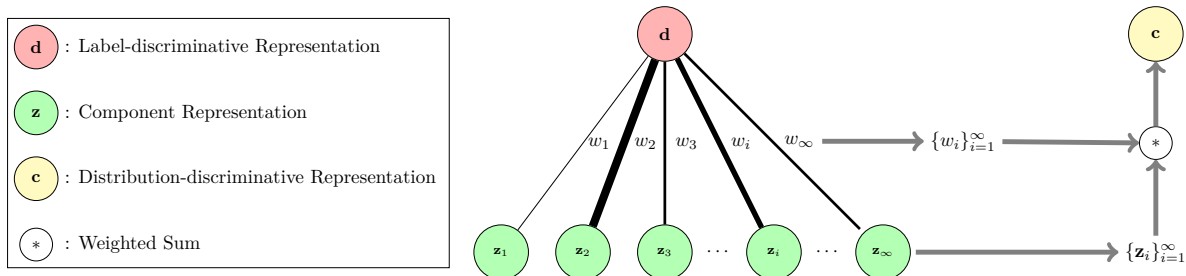

Figure 3: Constructing a distribution-discriminative representation. A thicker black line indicates a larger weight or vice versa. A distribution-discriminative representation consists of multiple component representations where a component representation less similar to the label-discriminative representation is given a higher weight.

to reduce the overlap information between $\mathcal{C}$ and $\mathcal{Y}$. Also, $\alpha$ is a difference coefficient controlling the trade-off between extracting label-related information from the original inputs and enlarging the difference between the label- and distribution-discriminative representations. A larger $\alpha$ will lead to less overlap information between $\mathcal{D}$ and $\mathcal{C}$ but less label information to be extracted, or vice versa.

Both Eq. (1) and Eq. (6) are learning principles for learning representations by finding a trade-off between learning label-related information and retaining less information of the original inputs. They are significantly different. Recall that Eq. (1) forces a network to explore the strongly label-related information by compression for learning the label-discriminative representations. Compared with Eq. (1), Eq. (6) considers a constraint $\mathcal{I}(\mathcal{D};\mathcal{C})$ which forces a network to explore the label-related information discarded by label-discriminative representations and use them to learn the distribution-discriminative representations. Therefore, based on the two learning principles, the label-related information of label- and distribution-discriminative representations is different and complementary.

To find a restriction for learning distribution-discriminative representations according to $\mathcal{I}(\mathcal{D};\mathcal{C})$, we quantify this mutual information term and have

$$\mathcal{I}(\mathcal{D};\mathcal{C}) = \mathbb{E}_{P(\mathcal{D})}\left[\mathrm{KL}\left(P(\mathcal{C}|\mathcal{D})\|P(\mathcal{C})\right)\right], \tag{7}$$

where $P(\mathcal{D})$, $P(\mathcal{C})$ and $P(\mathcal{C}|\mathcal{D})$ denote the respective probability distributions, and $\mathrm{KL}(\cdot|\cdot)$ represents the Kullback-Leibler (KL) divergence. However, measuring Eq. (7) is intractable because we cannot obtain an analytic expression for $P(\mathcal{C})$. According to the variational inference Gal & Ghahramani (2016), we can solve this problem by applying a tractable proposal distribution Bishop (2006) $Q(\mathcal{C})$ to approximate $P(\mathcal{C})$. We thus have

$$\mathcal{I}(\mathcal{D};\mathcal{C}) = \mathbb{E}_{P(\mathcal{D})}\left[\mathrm{KL}\left(P(\mathcal{C}|\mathcal{D})\|Q(\mathcal{C})\right)\right] - \mathrm{KL}\left(Q(\mathcal{C})\|P(\mathcal{C})\right) \leq \mathbb{E}_{P(\mathcal{D})}\left[\mathrm{KL}\left(P(\mathcal{C}|\mathcal{D})\|Q(\mathcal{C})\right)\right], \tag{8}$$

where the inequality is due to the nonnegative property of the KL divergence. According to Eq. (8), we know that the distribution-discriminative representations depend on the corresponding label-discriminative representations, and the distribution of distribution-discriminative representations $P(\mathcal{C}|\mathcal{D})$ should be close to the proposal distribution $Q(\mathcal{C})$. However, we have to decide $Q(\mathcal{C})$ according to prior knowledge which is expected to be similar to the unknown $P(\mathcal{C})$. Therefore, it is difficult to construct an explicit constraint, e.g., regularizer, for the distribution-discriminative representation learning due to the unknown proposal distribution $Q(\mathcal{C})$ in Eq. (8). The unknown proposal distribution drives us to explore an implicit constraint to ensure that a distribution-discriminative representation is complementary to its label-discriminative representation, i.e., a distribution-discriminative representation contains weakly label-related information discarded by the label-discriminative representation.

The three sources of mutual information $\mathcal{I}(\mathcal{C};\mathcal{Y})$, $\mathcal{I}(\mathcal{X};\mathcal{C})$ and $\mathcal{I}(\mathcal{D};\mathcal{C})$ in Eq. (6) can be modeled by a loss function, a network, and a constraint, respectively. Both the loss function and the network are not explicitly inferred from their mutual information terms. Specifically, the loss function ensures that the distribution-discriminative representations contain label-related information, and the network compresses the information

from inputs to learn their representations due to the down-sampling nature. Accordingly, based on $\mathcal{I}(\mathcal{D};\mathcal{C})$, we implicitly restrict that a distribution-discriminative representation is different from its corresponding label-discriminative representation in the learning process. This implicit constraint forces the network to explore weakly label-related information since the strongly label-related information has been explored by label-discriminative representations $\mathcal{D}$. Without this constraint, networks will explore the same strongly label-related information in $\mathcal{D}$ from inputs to learn representations according to $\max \mathcal{I}(\mathcal{C};\mathcal{Y}) - \beta_{\mathcal{C}}\mathcal{I}(\mathcal{X};\mathcal{C})$. Because $\mathcal{I}(\mathcal{C};\mathcal{Y})$, $\mathcal{I}(\mathcal{X};\mathcal{C})$ and $\mathcal{I}(\mathcal{D};\mathcal{C})$ in Eq. (6) are implicitly modeled, it is unnecessary to explicitly set the hyper-parameters $\beta_{\mathcal{C}}$ and $\alpha$ in Eq. (8).

**Discussion:** The information bottleneck principle offers a principled approach to encourage the learning of more abstract and invariant representations from input data for a given task, which forces networks to capture the underlying structures and dependencies in the data that are relevant to this task. Accordingly, using different constraints, we can obtain the learning principles for label- and distribution-discriminative representations, as shown in Eq. (1) and Eq. (6), respectively. Specifically, label-discriminative representations aim to capture the strongly label-related information, while distribution-discriminative representations aim to capture the remaining weakly label-related information. However, the information bottleneck principle finds a trade-off between compression and loss of information. Therefore, exploring all the label-related information is impossible, and some essential details in the data may be discarded.

### 3.3 Learning Auxiliary Networks

Following the information bottleneck principle in Eq. (6), we apply an auxiliary network $f_\theta$ with an implicit constraint to learn a complementary distribution-discriminative representation for its corresponding label-discriminative representation, i.e.,

$$\mathbf{c} = f_\theta(\mathbf{x}, \mathbf{d}). \tag{9}$$

Directly modeling $f_\theta(\mathbf{x}, \mathbf{d})$ by a network cannot ensure that $\mathbf{d}$ and $\mathbf{c}$ are different, i.e., ensuring $\mathbf{d}$ and $\mathbf{c}$ are learned from strongly and weakly label-related information, respectively. This is because it is difficult to design an implicit constraint for $f_\theta(\mathbf{x}, \mathbf{d})$ due to the unknown proposal distribution $Q(\mathcal{C})$ in Eq. (8). We develop an indirect modeling method by an implicit constraint. The basic idea is to decompose the distribution-discriminative representation $\mathbf{c}$ into multiple *component representations* $\{\mathbf{z}_1, \ldots, \mathbf{z}_\infty\}$ where a component representation $\mathbf{z}_i$ less similar to the label-discriminative representation $\mathbf{d}$ is assigned with a higher weight $w_i$, as shown in Figure 3.

#### 3.3.1 Decompose Distribution-Discriminative Representations

We firstly decompose Eq. (9) into a linear combination,

$$\mathbf{c} = \sum_{i=1}^{\infty} w_i \mathbf{z}_i, \tag{10}$$

We assume $\mathbf{z}_i$ is drawn from the Gaussian distribution $\mathcal{N}(\mu_{\mathcal{Z}}, \Sigma_{\mathcal{Z}})$ without loss of generality. Inspired by the determinant point processes Kulesza & Taskar (2012); Anari et al. (2016) selecting diverse samples according to the kernel-based distance Kulesza & Taskar (2011), we construct $\mathbf{c}$ by integrating diverse component representations $\{\mathbf{z}_1, \ldots, \mathbf{z}_\infty\}$ where an component representation $\mathbf{z}$ less similar with the corresponding label-discriminative representation $\mathbf{d}$ has a higher weight. Simulating the idea of the attention mechanism Vaswani et al. (2017), the inner product is adopted as the similarity metric. We thus define the weight $w_i$ as

$$w_i = 1 - \epsilon \mathbf{z}_i^T \mathbf{d}. \tag{11}$$

where $\epsilon$ is a small perturbation coefficient to ensure that the weight is positive. Substituting Eq. (11) into Eq. (10), we obtain an implicit constraint on $\mathbf{c}$,

$$\mathbf{c} = \mathbb{E}_{\mathbf{z}}\left[\mathbf{z} - \epsilon \mathbf{z}^T \mathbf{d}\mathbf{z}\right] = \mathbb{E}_{\mathbf{z}}[\mathbf{z}] - \epsilon \mathbb{E}_{\mathbf{z}}\left[(\mathbf{z} - \mu_{\mathcal{Z}})^T \mathbf{d}(\mathbf{z} - \mu_{\mathcal{Z}})\right] - \epsilon \mu_{\mathcal{Z}}^T \mathbf{d}\mu_{\mathcal{Z}} = \mu_{\mathcal{Z}} - \epsilon\left(\Sigma_{\mathcal{Z}}\mathbf{d} + \mu_{\mathcal{Z}}^T \mathbf{d}\mu_{\mathcal{Z}}\right). \tag{12}$$

### 3.3.2 Estimate Distribution-Discriminative Representations

According to Eq. (12), we can estimate a distribution-discriminative representation $\mathbf{c}$ by estimating the expectation $\mu_{\mathcal{Z}}$ and the covariance matrix $\Sigma_{\mathcal{Z}}$. Following Bayesian neural networks Gal & Ghahramani (2016); Ardywibowo et al. (2022), we estimate $\mu_{\mathcal{Z}}$ by applying an *component network $z_\theta$* which maps an input $\mathbf{x}$ to the component representation expectation. i.e.,

$$z_\theta(\mathbf{x}) = \mu_{\mathcal{Z}}. \tag{13}$$

Note that $f_\theta$ and $z_\theta$ share the same network backbone and parameter $\theta$ because we indirectly construct $f_\theta$ by Eq. (9), Eq. (12) and Eq. (13). Accordingly, the relationship between $f_\theta$ and $z_\theta$ is

$$f_\theta(\mathbf{x}, \mathbf{d}) = z_\theta(\mathbf{x}) - \epsilon \left( \Sigma_{\mathcal{Z}} \mathbf{d} + z_\theta(\mathbf{x})^T \mathbf{d} z_\theta(\mathbf{x}) \right), \tag{14}$$

which is the implicit constraint for learning distribution-discriminative representations, i.e., constructing a distribution-discriminative representation by integrating multiple component representations differing from its corresponding label-discriminative representations. The indirect construction method considers the implicit constraint to ensure that the distribution-discriminative representations are different from the corresponding label-discriminative representations, which ensures that label- and distribution-discriminative representations are learned from strongly and weakly label-related information, respectively.

However, the covariance matrix $\Sigma_{\mathcal{Z}}$ is still unknown. We can define the covariance matrix according to our prior knowledge. This is because $\Sigma_{\mathcal{Z}}$ which represents the dispersion degree of component representation $\mathbf{z}$ does not play a decisive role in learning distribution-discriminative representations. Since the covariance matrix $\Sigma_{\mathcal{D}}$ of ID samples can be easily estimated by the pretrained network $g_\theta$. We thus assume $\Sigma_{\mathcal{Z}} = \Sigma_{\mathcal{D}}$ without loss of generality.

### 3.3.3 Objective Function

Following the learning principle of label-discriminative representations, we apply the cross-entropy loss to model $\mathcal{I}(\mathcal{C}; \mathcal{Y})$ and assume $h(\cdot, \cdot)$ is the softmax function. Accordingly, the objective function for learning distribution-discriminative representations is,

$$\min_\theta -\mathbb{E}_{(\mathbf{x}, y) \sim P(\mathcal{X}, \mathcal{Y})} \log h(\mathbf{c}, y),$$
$$\text{s.t.} \quad \mathbf{c} = f_\theta(\mathbf{x}, \mathbf{d}), \mathbf{d} = g_\phi(\mathbf{x}). \tag{15}$$

The parameter $\phi$ is fixed for learning the parameter $\theta$ in $f_\theta$ because $g_\phi$ is a pretrained network. Based on Monte Carlo Wu & Gleich (2019), we apply the stochastic gradient descent optimization algorithm Shalev-Shwartz & Ben-David (2014) to estimate the gradient of Eq. (15), where the batch size is $B$.

### 3.4 Calculating Out-of-distribution Scores

For an ID sample, both its label- and distribution-discriminative representations contain information pointing to the same label. For an OOD sample with high-confidence prediction, its label-discriminative representation is sensitive to a label, while its distribution-discriminative representation is sensitive to other labels or even none of any labels. Therefore, the labeling information in the label- and distribution-discriminative representations are complementary for ID samples but inconsistent for OOD samples. Therefore, we can detect OOD samples by combining these two representations. We obtain a softmax output of label $y$ for input $\mathbf{x}$ by simply averaging the softmax outputs of the two representations, i.e.,

$$O(\mathbf{x}, y) = \frac{h(\mathbf{c}, y) + h(\mathbf{d}, y)}{2} = \frac{h(f_\theta(\mathbf{x}, g_\phi(\mathbf{x})), y) + h(g_\phi(\mathbf{x}), y)}{2}. \tag{16}$$

We classify ID samples according to the softmax outputs $\{O(\mathbf{x}, 1), \ldots, O(\mathbf{x}, K)\}$. Following the baseline method Hendrycks & Gimpel (2017) of detecting OOD samples which uses the confidence as the OOD score, we calculate the OOD score for input $\mathbf{x}$ by

$$\mathcal{S}(\mathbf{x}) = \max_{y \in [1, K]} O(\mathbf{x}, y), \tag{17}$$

---

**Algorithm 1** Dual Representation Learning (DRL)

---

1: **Input:** pretrained network $g_\phi$, perturbation coefficient $\epsilon$, covariance $\Sigma_\mathcal{D}$, batch size $B$
2: **while** no convergence **do**
3:     Sample $\{(\mathbf{x}_1, \mathbf{y}_1), \ldots, (\mathbf{x}_B, \mathbf{y}_B)\}$ from $P(\mathcal{X}, \mathcal{Y})$
4:     Receive $\mathbf{d}_i = g_\phi(\mathbf{x}_i), \forall i \in [B]$
5:     Calculate $\mathbf{c}_i = f_\theta(\mathbf{x}_i, \mathbf{d}_i), \forall i \in [B]$
6:     Estimate the objective function:

$$\widetilde{\mathcal{L}}(\theta) = -\frac{1}{B} \sum_{i=1}^{B} \log h(\mathbf{c}_i, y_i)$$

7:     Obtain gradients $\nabla_\theta \widetilde{\mathcal{L}}(\theta)$ to update parameters $\theta$
8: **end while**
9: Calculate out-of-distribution score:

$$\mathcal{S}(\mathbf{x}) = \max_{y \in [1, K]} \left( h(\mathbf{c}, y) + h(\mathbf{d}, y) \right) / 2$$

10: **Output:** $\mathcal{S}(\mathbf{x})$

---

where an ID sample is expected to have a higher score, whereas an OOD sample is expected to have a lower score. The pseudo-code of the DRL training procedure is summarized in Algorithm 1.

## 4 Experiments

In this section, we demonstrate the effectiveness of the proposed DRL method[1]. We compare DRL with post-hoc, confidence enhancement and ensemble methods. Furthermore, we analyze the effect of the hyper-parameters and network backbones in DRL, run a set of ablation study experiments, and show the sensitivity of labeling information of label- and distribution-discriminative representations.

### 4.1 Setups

We adopt the ResNet18 architecture He et al. (2016) for all the networks in the experiments and implement it in PyTorch. The learning rate starts at 0.1 and is divided by 10 after 100 and 150 epochs in the training phase, and all networks are trained for 200 epochs with 128 samples per mini-batch. If not specified, we set $\epsilon = 0.001$ and adopt the same network architecture for pretrained and auxiliary networks in the proposed method. For the pretrained network, we train it with a cross-entropy loss on an ID dataset. For learning the auxiliary network, we adopt $\Sigma_\mathcal{Z} = \Sigma_\mathcal{D}$ for constructing distribution-discriminative representations if not specified. Recall that $\Sigma_\mathcal{D}$ is the covariance matrix of label-discriminative representations of ID samples from the pretrained network.

We adopt CIFAR10 Krizhevsky (2009) and Mini-Imagenet Deng et al. (2009) as ID datasets to train neural networks. The numbers of classes of the two ID datasets are 10 and 100, respectively. The resolution ratios are $32 \times 32$ and $224 \times 224$, respectively. For data augmentation methods, we apply random crop and random horizontal flip for CIFAR10 and resizing and random crop for Mini-Imagenet. We adopt CIFAR100 Krizhevsky (2009), CUB200 Wah et al. (2011), StanfordDogs120 Khosla et al. (2011), OxfordPets37 Parkhi et al. (2012), Oxfordflowers102 Nilsback & Zisserman (2006), Caltech256 Griffin et al. (2006), DTD47 Cimpoi et al. (2014), and COCO Lin et al. (2014) as OOD datasets to evaluate the OOD detection performance in the test phase.

In the test phase, each method calculates an OOD score for each test sample. The score gap between ID and OOD samples is expected to be large, which indicates we can clearly separate the two kinds of samples. Accordingly, to evaluate the detection performance of OOD samples, we adopt the following three metrics:

---

[1]The source codes are at: `https://github.com/Lawliet-zzl/DRL`

Table 1: OOD detection performance of six post-hoc methods and DRL. All the reported values are averaged AUROC over five trials. The subscript values denote the standard deviation. 'Ave.' represents the averaged AUROC across all eight OOD datasets. Boldface values represent the relatively better detection performance.

| In-dist | Out-of-dist | Baseline | ODIN | ED | MDD | RA | GEM | DRL |
|---------|-------------|----------|------|-----|-----|-----|-----|-----|
| CIFAR10 | CIFAR100 | $87.0\pm_{0.2}$ | $86.0\pm_{0.0}$ | $86.5\pm_{0.0}$ | $85.8\pm_{0.1}$ | $87.4\pm_{0.1}$ | $87.1\pm_{0.1}$ | $\mathbf{89.5}\pm_{0.1}$ |
| | CUB200 | $61.5\pm_{0.4}$ | $56.0\pm_{0.0}$ | $57.1\pm_{0.0}$ | $\mathbf{66.5}\pm_{0.1}$ | $62.9\pm_{0.1}$ | $62.3\pm_{0.2}$ | $63.7\pm_{0.4}$ |
| | StanfordDogs120 | $68.0\pm_{0.7}$ | $64.9\pm_{0.0}$ | $66.9\pm_{0.0}$ | $72.4\pm_{0.1}$ | $69.4\pm_{0.0}$ | $71.3\pm_{0.1}$ | $\mathbf{73.1}\pm_{0.1}$ |
| | OxfordPets37 | $62.9\pm_{1.7}$ | $60.2\pm_{0.0}$ | $61.6\pm_{0.0}$ | $65.3\pm_{0.1}$ | $63.0\pm_{0.2}$ | $66.3\pm_{0.0}$ | $\mathbf{67.8}\pm_{0.2}$ |
| | Oxfordflowers102 | $88.1\pm_{0.6}$ | $87.5\pm_{0.0}$ | $87.3\pm_{0.0}$ | $88.9\pm_{0.0}$ | $88.4\pm_{0.0}$ | $90.4\pm_{0.1}$ | $\mathbf{91.2}\pm_{0.1}$ |
| | Caltech256 | $86.0\pm_{0.2}$ | $86.4\pm_{0.0}$ | $85.6\pm_{0.0}$ | $85.2\pm_{0.0}$ | $86.5\pm_{0.0}$ | $84.1\pm_{0.1}$ | $\mathbf{88.0}\pm_{0.1}$ |
| | DTD47 | $89.4\pm_{1.5}$ | $91.0\pm_{0.0}$ | $90.6\pm_{0.0}$ | $89.2\pm_{0.0}$ | $89.0\pm_{0.0}$ | $88.4\pm_{0.0}$ | $\mathbf{92.6}\pm_{0.1}$ |
| | COCO | $87.3\pm_{0.1}$ | $87.8\pm_{0.0}$ | $87.6\pm_{0.0}$ | $86.6\pm_{0.0}$ | $87.4\pm_{0.1}$ | $85.2\pm_{0.0}$ | $\mathbf{89.2}\pm_{0.1}$ |
| | Ave. | 78.7 | 78.7 | 77.9 | 79.9 | 79.2 | 79.4 | **81.9** |
| Mini-Imagenet | CIFAR100 | $84.7\pm_{0.2}$ | $85.8\pm_{0.0}$ | $84.0\pm_{0.0}$ | $84.1\pm_{0.1}$ | $84.6\pm_{0.1}$ | $85.3\pm_{0.1}$ | $\mathbf{86.4}\pm_{0.1}$ |
| | CUB200 | $71.8\pm_{0.4}$ | $71.5\pm_{0.0}$ | $70.1\pm_{0.0}$ | $\mathbf{77.1}\pm_{0.1}$ | $72.1\pm_{0.1}$ | $74.4\pm_{0.1}$ | $73.8\pm_{0.4}$ |
| | StanfordDogs120 | $65.6\pm_{0.7}$ | $63.6\pm_{0.0}$ | $64.0\pm_{0.0}$ | $62.0\pm_{0.1}$ | $63.5\pm_{0.1}$ | $65.7\pm_{0.1}$ | $\mathbf{67.3}\pm_{0.1}$ |
| | OxfordPets37 | $70.4\pm_{1.7}$ | $68.6\pm_{0.0}$ | $69.6\pm_{0.0}$ | $64.5\pm_{0.1}$ | $67.4\pm_{0.1}$ | $68.9\pm_{0.1}$ | $\mathbf{72.2}\pm_{0.2}$ |
| | Oxfordflowers102 | $79.5\pm_{0.6}$ | $80.4\pm_{0.0}$ | $78.0\pm_{0.0}$ | $76.5\pm_{0.0}$ | $77.6\pm_{0.1}$ | $79.8\pm_{0.0}$ | $\mathbf{81.4}\pm_{0.1}$ |
| | Caltech256 | $78.2\pm_{0.2}$ | $79.8\pm_{0.0}$ | $78.5\pm_{0.0}$ | $71.2\pm_{0.0}$ | $75.4\pm_{0.0}$ | $78.3\pm_{0.0}$ | $\mathbf{79.8}\pm_{0.1}$ |
| | DTD47 | $72.8\pm_{1.5}$ | $73.9\pm_{0.0}$ | $71.9\pm_{0.0}$ | $\mathbf{76.1}\pm_{0.0}$ | $72.6\pm_{0.0}$ | $78.6\pm_{0.0}$ | $74.5\pm_{0.1}$ |
| | COCO | $78.9\pm_{0.1}$ | $79.2\pm_{0.0}$ | $78.0\pm_{0.0}$ | $73.9\pm_{0.0}$ | $73.9\pm_{0.0}$ | $76.1\pm_{0.0}$ | $\mathbf{79.9}\pm_{0.1}$ |
| | Ave. | 75.2 | 75.3 | 74.2 | 73.1 | 75.1 | 75.9 | **76.9** |

the area under the receiver operating characteristic curve (AUROC) Davis & Goadrich (2006), the true negative rate at 95% (FPR95) Liang et al. (2018) and the detection accuracy (Detection) Liang et al. (2018). A higher AUROC score and a lower FPR95 and Detection score indicate better detection performance. Specifically, AUROC indicates the probability that a higher score is assigned to an ID sample than an OOD sample. FPR95 indicates the probability that an OOD sample is declared to be an ID sample when the true positive rate is 95%. Detection indicates the misclassification probability when the true positive rate is 95%. In addition to OOD detection performance, we also evaluate the classification Accuracy and Expected Calibration Error (ECE) Guo et al. (2017). ECE uses the difference in expectation between confidence and accuracy to measure calibration. A network is considered calibrated if its predictive confidence aligns with its misclassification rate. In our experiments, the number of bins in ECE is set to 20.

## 4.2 Comparison Results

### 4.2.1 Comparison with Post-hoc Methods

We compare DRL with six post-hoc methods that train an OOD detector according to the outputs from a pretrained network. DRL does not modify the pretrained network, which is the same as the post-hoc methods. Rather than training an OOD detector, DRL improves OOD sensitivity by training an auxiliary network for the pretrained network to explore OOD-sensitive information. All the compared methods are based on pretrained networks. The pretrained networks used in the post-hoc methods are the same as that in DRL.

Table 2: OOD detection performance of seven confidence enhancement methods and DRL. Each value is averaged across all eight OOD datasets. The symbol ↑ indicates a larger value is better, and the symbol ↓ indicates a lower value is better. The boldface value represents the relatively better detection performance.

| Dataset | Metric | JCL | CSI | SSL | DeConf-C | MOS | KNN+ | CIDER | DRL |
|---------|--------|-----|-----|-----|----------|-----|------|-------|-----|
| CIFAR10 | AUROC ↑ | 77.5 | 79.2 | 78.1 | 78.4 | 77.8 | 79.2 | 80.2 | **81.9** |
| | FPR(95) ↓ | 73.9 | 67.8 | **62.1** | 64.8 | 68.2 | 67.8 | 66.9 | 65.0 |
| | Detection ↓ | 26.7 | 24.6 | 26.6 | 25.9 | 27.5 | 25.3 | 24.6 | **22.5** |
| Mini-Imagenet | AUROC ↑ | 72.8 | 75.2 | 75.6 | 75.3 | 75.3 | 74.5 | 75.4 | **76.9** |
| | FPR(95) ↓ | 86.5 | 85.9 | **82.5** | 85.8 | 86.3 | 86.2 | 85.3 | 83.2 |
| | Detection ↓ | 32.0 | 29.9 | 28.9 | 29.7 | 29.1 | 29.6 | 29.0 | **28.3** |

The considered post-hoc methods include the Baseline Hendrycks & Gimpel (2017), ODIN Liang et al. (2018), Energy-based Detector (ED) Liu et al. (2020), Mahalanobis Distance-based Detector (MDD) Lee et al. (2018b), Rectified Activations (RA) Sun et al. (2021), and Gaussian mixture based Energy Measurement Morteza & Li (2022). For a fair comparison, we add the scores from different layers without training a logistic regression on a validation OOD dataset in MDD because the other compared methods do not access OOD samples in the training phase. For the other comparison methods, we follow the same setups as their original ones.

The comparison results are summarized in Table 1. When the training ID dataset is CIFAR10, DRL achieves the best OOD detection performance on seven of the eight OOD datasets. Furthermore, when the training ID dataset is Mini-Imagenet, a more complex dataset with more classes and higher resolution, DRL achieves the best detection performance on six of the eight OOD datasets and obtains the second-best detection performance on the rest two OOD datasets. Overall, DRL achieves the best averaged OOD detection performance across diverse OOD datasets, which indicates that DRL outperforms the state-of-the-art post-hoc methods. The reason is twofold. The post-hoc methods based on pretrained networks can only access the label-discriminative representations that are sensitive to labeling. However, DRL trains an auxiliary network for a given pretrained network to explore the complementary distribution-discriminative representations that are sensitive to OOD samples.

### 4.2.2 Comparison with Confidence Enhancement Methods

We compare DRL with the state-of-the-art confidence enhancement methods in terms of AUROC, FPR(95) and Detection. Confidence enhancement methods retrain a pretrained network to improve the OOD sensitivity by modifying loss functions and training procedures. The adopted methods include JCL Lee et al. (2018a), CSI Tack et al. (2020), SSL Hendrycks et al. (2019b), DeConf-C Hsu et al. (2020), MOS Huang & Li (2021), KNN+ Sun et al. (2022), and Compactness and Dispersion Regularized learning Ming et al. (2023). The settings of all the comparison methods follow the original ones. For a fair comparison, following the grouping method in MOS, we apply K-Means clustering on feature representations to divide the training ID datasets into groups with similar concepts.

The comparison results are summarized in Table 2. We observe that DRL obtains significant improvement (4.64% on CIFAR10 and 2.04% on Mini-Imagenet) over the other state-of-the-art confidence enhancement methods in terms of AUROC. Likewise, DRL achieves significantly improved performance (13.90% on CIFAR10 and 5.12% on Mini-Imagenet) in terms of Detection. Although DRL does not obtain the best results in terms of FPR(95), its performance is close to the best one with a narrow gap (2.9 on CIFAR10 and 0.7 on Mini-Imagenet). Overall, the proposed DRL method outperforms the state-of-the-art confidence enhancement methods on both datasets in terms of AUROC and Detection. The results demonstrate that exploring distribution-discriminative representations from weakly label-related information and integrating label- and distribution-discriminative representations form an effective mechanism to detect OOD samples. It is because

Table 3: OOD detection performance on near and far OOD samples in terms of AUROC. The boldface value represents the relatively better detection performance.

| Dataset | Metric | JCL | CSI | SSL | DeConf-C | MOS | KNN+ | CIDER | DRL |
|---------|--------|-----|-----|-----|----------|-----|------|-------|-----|
| CIFAR10 | CIFAR100 (Near) | 84.3 | 87.6 | 88.2 | 89.2 | 88.4 | 85.6 | 89.1 | **89.5** |
| | CUB200 (Near) | 60.9 | 61.2 | 62.1 | 62.4 | 60.9 | 62.9 | 63.0 | **63.7** |
| | Oxfordflowers102 (Far) | 84.6 | 85.4 | 90.4 | 90.1 | 88.7 | 89.6 | 90.5 | **91.2** |
| | DTD47 (Far) | 88.6 | 88.6 | 90.5 | 91.2 | 90.3 | 90.3 | 91.6 | **92.6** |

the distribution-discriminative representations coupled with the corresponding label-discriminative representations contain more label-related information than any of them, which reduces the prediction confidence for an OOD sample owning minimum label-related information and enhances the prediction confidence for an ID sample owning all the labeling information. It enlarges the confidence gap between ID and OOD samples. Therefore, DRL improves OOD detection performance by leveraging more label-related information, which indicates that DRL may fail if the training ID samples contain little label-related information with numerous label-unrelated information. This is because training on these ID samples causes no remaining label-related information that the auxiliary network of DRL can explore.

Following the setups of Winkens et al. Winkens et al. (2020), we also evaluate the OOD detection performance on near and far samples. Near and far samples represent the out-of-distribution samples slightly and significantly different from ID samples, respectively. For the ID dataset CIFAR10, we adopt CIFAR100 and CUB200 as near OOD datasets because they contain some classes having similar semantics to that of CIFAR10. Furthermore, we adopt Oxfordflowers102 and DTD47 as far OOD datasets because the classes of the two datasets are vastly different from that of CIFAR10. The experimental results are summarized in Table 3. All the considered methods can effectively detect far OOD samples due to the large semantic gap of classes between ID and OOD datasets. Furthermore, DRL achieves 2.62% improvement over the compared methods. Conversely, it is difficult to distinguish near OOD samples from ID samples since the performance of each method is lower than that on far OOD samples. However, DRL still achieves the best detection performance and obtains more significant improvement, i.e., 2.94%. This is because DRL improves OOD sensitivity by exploring more label-related information from original inputs. Therefore, DRL can leverage more details to describe ID samples, which can more effectively differentiate from OOD samples with different information.

### 4.2.3 Comparison with Ensemble Methods

We compare DRL with ensemble methods that integrate independent networks with different initialization parameters. Specifically, we apply the traditional deep ensemble method which combines the output predictions by calculating the average. Furthermore, we incorporate the adversarial samples into the deep ensemble method Guyon et al. (2017). Note that the traditional ensemble method degrades into the baseline method when the number of networks equals one. All the methods utilize multiple networks to explore more information from the ID samples to improve OOD sensitivity.

The comparison results are summarized in Figure 4. For the two ensemble methods, the OOD detection increases as the number of networks increases on both CIFAR10 and Mini-Imagenet datasets. Furthermore, considering adversarial samples for the deep ensemble method leads to improved and declined performance on CIFAR10 and Mini-Imagenet, respectively. Overall, DRL achieves the best detection performance on both datasets, and the performance of the two ensemble methods is close to that of DRL when the number of networks is five. Therefore, DRL only containing two dependent networks outperforms the ensemble methods containing more independent networks. The results verify that the proposed DRL method differs from the ensemble method substantially. Further, DRL combines label- and distribution-discriminative representations to fetch the OOD-sensitive information from training ID samples.

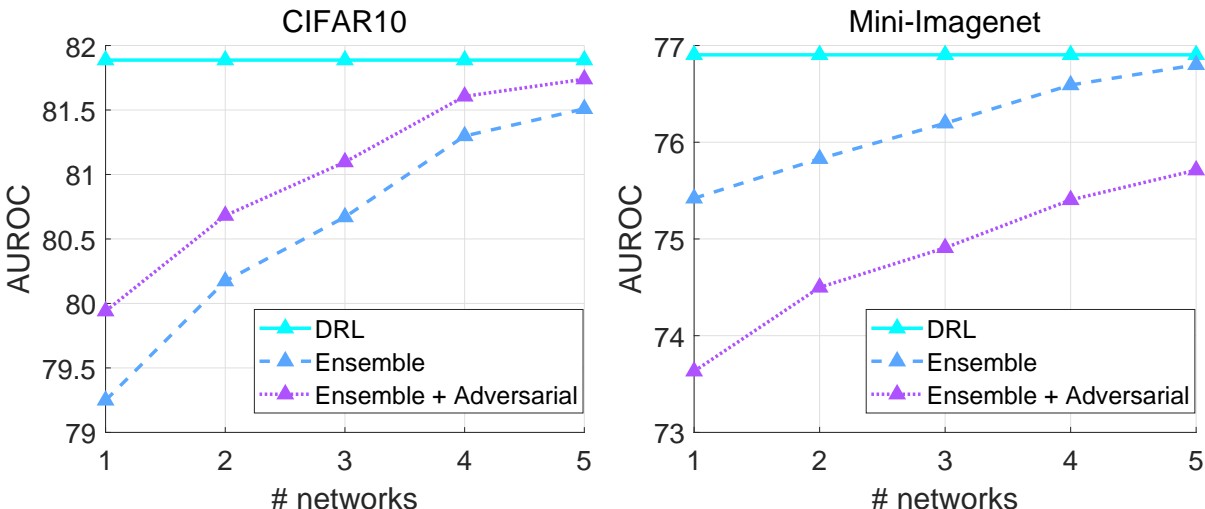

Figure 4: OOD detection performance of two ensemble methods and DRL. The ensemble networks gradually increase independent pretrained networks with randomized initialization, and DRL only owns two depent networks, i.e, a pretrained and an auxiliary network. Each point indicates the average AUROC across all eight OOD datasets.

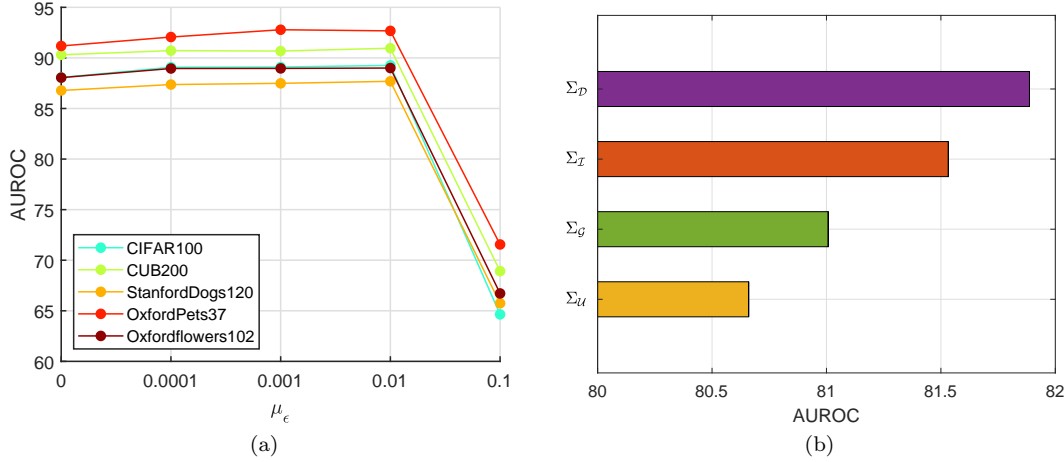

Figure 5: (a) Effect of the perturbation coefficient $\epsilon$ on CIFAR10. Each curve represents the detection performance on an OOD dataset, and each point indicates the AUROC for the corresponding perturbation coefficient expectation $\epsilon$; (b) Effect of the covariance matrix $\Sigma_{\mathcal{Z}}$ on CIFAR10. Each bar indicates the average AUROC across all eight OOD datasets.

## 4.3 Parameter Analysis

### 4.3.1 Effect of perturbation coefficient $\epsilon$

According to Eq. (11), we adopt a small perturbation coefficient $\epsilon$ to ensure positive weight $w$. We also empirically verify that a small perturbation coefficient $\epsilon$ is more suitable for the proposed DRL method. We select the value of the perturbation coefficient expectation $\epsilon$ in $\{0, 0.0001, 0.001, 0.01, 0.1\}$. Note that DRL degenerates to deep ensemble methods containing two independent networks when $\epsilon = 0$.

The experimental results are shown in Figure 5a. We observe that increasing $\epsilon$ can gradually improve the detection performance, although the effect is drastically reduced when $\epsilon$ is sufficiently large ($\epsilon > 0.01$). Therefore, DRL prefers a perturbation coefficient that is small but larger than zero. According to the

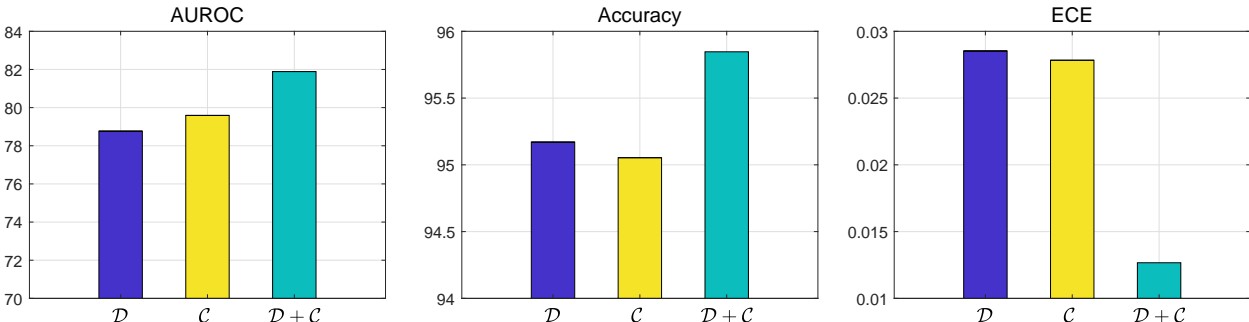

Figure 6: Results of the ablation study on CIFAR10. The three subfigures report the results of AUROC, Accuracy and ECE, respectively. $\mathcal{D}$ indicates the label-discriminative representations from a pretrained network. $\mathcal{C}$ indicates the complementary distribution-discriminative representations from an auxiliary network. $\mathcal{D} + \mathcal{C}$ indicates the combination of label- and distribution-discriminative representations, i.e., the proposed DRL method.

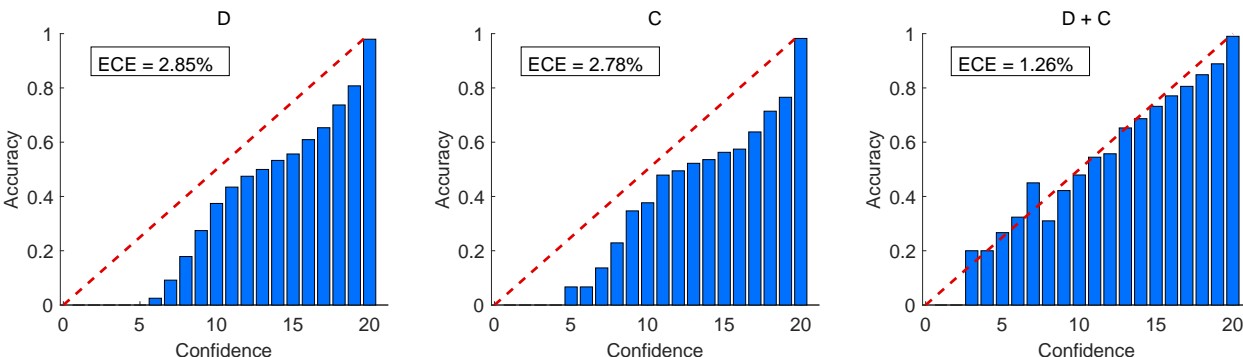

Figure 7: Calibration results on CIFAR10. The confidence is equally divided into 20 intervals, and each bar represents the expected accuracy of samples whose confidence values are in the same interval. The red dotted diagonal indicates the perfect calibration.

weight $w$ measuring the dissimilarity between a label-discriminative representation $\mathbf{d}$ and a component representation $\mathbf{z}$, $\epsilon$ is applied to ensure that $w$ is positive. However, a large $\epsilon$ cannot satisfy the criteria. From another point of view, the expression of the distribution-discriminative representation $\mathbf{c}$ in Eq. (12) is similar to adversarial samples Goodfellow et al. (2015) where the coefficient $\epsilon$ affects the portion of the perturbation $\Sigma_{\mathcal{Z}}\mathbf{d} + \mu_{\mathcal{Z}}^T\mathbf{d}\mu_{\mathcal{Z}}$ on $\mu_{\mathcal{Z}}$. According to the idea of adversarial learning, a small coefficient is sufficient to alter the predicted label of a test sample. Similarly, in DRL, a small $\epsilon$ is sufficient to construct a distribution-discriminative representation $\mathbf{c}$ significantly differing from the corresponding label-discriminative representation, and a large $\epsilon$ can cause an unstable distribution-discriminative representation which cannot capture the weakly label-related information from the training ID samples.

### 4.3.2   Effect of covariance matrix $\Sigma_{\mathcal{Z}}$

We empirically verify that DRL is not sensitive to the choice of covariance matrix $\Sigma_{\mathcal{Z}}$, and it is more suitable to adopt the covariance matrix $\Sigma_{\mathcal{D}}$ estimated from label-discriminative representations to establish the implicit constraint Eq. (14) for learning the corresponding distribution-discriminative representations, i.e., $\Sigma_{\mathcal{Z}} = \Sigma_{\mathcal{D}}$. We compare $\Sigma_{\mathcal{D}}$ with different choices of covariance matrix $\Sigma_{\mathcal{Z}}$, including the identity matrix $\Sigma_{\mathcal{I}}$, the Gaussian random matrix $\Sigma_{\mathcal{G}}$, and the uniform random matrix $\Sigma_{\mathcal{U}}$.

The experimental results are shown in Figure 5b. Overall, the performance gap ($[80.7, 81.9]$) among the four different covariance matrices is not significant. This indicates that the covariance matrix $\Sigma_{\mathcal{Z}}$ does not play a decisive role in learning distribution-discriminative representations, and DRL is not sensitive to the

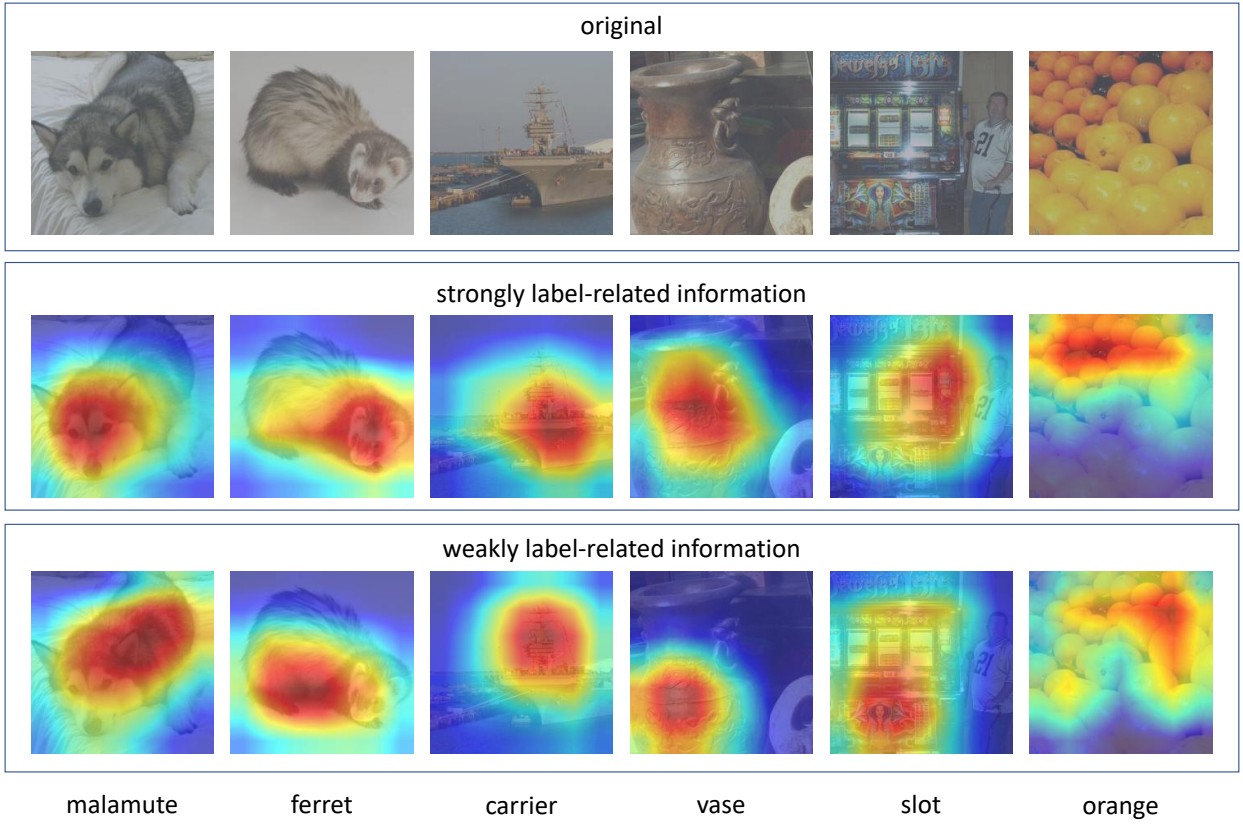

Figure 8: Heat maps of Grad-CAM for label- and distribution-discriminative representations on Mini-Imagenet. Red regions correspond to high scores for class, while blue regions correspond to low scores. The first row represents the original images, the second row represents their strongly label-related information for learning label-discriminative representations, and the third row represents their weakly label-related information for learning distribution-discriminative representations. The figure is best viewed in color.

choice of $\Sigma_{\mathcal{Z}}$. Specifically, the two random matrices $\Sigma_{\mathcal{G}}$ and $\Sigma_{\mathcal{U}}$ achieve relatively poor performance, while the identity matrix achieves relatively better performance. We expect that this is because networks tend to decouple the features within an output representation. Therefore, the identity matrix, which assumes all features are independent, is more appropriate than the two random matrices assuming random feature relationships. Adopting $\Sigma_{\mathcal{D}}$ achieves slightly better performance than $\Sigma_{\mathcal{I}}$. This is because $\Sigma_{\mathcal{D}}$ is obtained from the pretrained network, and the pretrained and auxiliary networks own the same network structure and are trained on the same ID dataset. Therefore, $\Sigma_{\mathcal{D}}$ is the most appropriate estimation for $\Sigma_{\mathcal{Z}}$. Note that the covariance matrix only represents the dispersion degree of representations. Accordingly, assuming the pretrained and auxiliary networks own the same covariance matrix does not indicate their output representations satisfy the same unknown distribution.

### 4.4 Auxiliary Network Backbone Analysis

To verify the effect of auxiliary network backbones, for the pretrained network with ResNet18 backbone He et al. (2016), we adopt Lightweight Network (LN), VGG19 Szegedy et al. (2015) and DenseNet100 Huang et al. (2017) for its corresponding auxiliary network. LN is a shallow multi-layer perceptron which is fully-connected and has two hidden layers of 128 ReLU units. The experimental results on CIFAR10 are shown in Table 4. LN obtains the worst detection performance since the shallow network merely explores limited weakly label-related information. Conversely, the rest three network architectures that are more powerful achieve better results. Specifically, DenseNet100 with the most significant number of parameters achieves

Table 4: Effect of auxiliary network backbones on CIFAR10. Each value is averaged across all eight OOD datasets. The boldface value represents the relatively better detection performance.

| LN | VGG19 | DenseNet100 | ResNet18 |
|------|-------|-------------|----------|
| 80.0 | 81.2 | **82.7** | 81.9 |

the best results. Accordingly, the auxiliary network of DRL can adapt to different network architectures, i.e., the networks for exploring strongly and weakly label-related information could have different network architectures. Furthermore, the detection performance depends on the power of the adopted network architecture.

## 4.5 Ablation Study

We run a set of ablation study experiments to verify that label- and distribution-discriminative representations are indispensable for improving OOD detection performance. Recall that DRL contains a pretrained network and an auxiliary network where the two networks learn label-discriminative representations ($\mathcal{D}$) and distribution-discriminative representations ($\mathcal{C}$), respectively, and apply the combination ($\mathcal{D} + \mathcal{C}$) of the two representations by Eq. (17) to detect OOD samples. Accordingly, we compare the combination with the two different representations in terms of AUROC, Accuracy and ECE.

The results are presented in Figure 6 and Figure 7. They show that exploiting only the label-discriminative representations has similar performance to exploiting only the distribution-discriminative representations. Specifically, the distribution-discriminative representations achieve slightly better performance in detecting OOD samples but slightly worse performance in classifying ID samples than the label-discriminative representations. The main reason is that the distribution-discriminative representation of an ID sample contains information that is weakly related to its labeling, and the weakly label-related information is more sensitive to OOD samples than the strongly-related information in the label-discriminative representations. From Figure 7, we observe that the pretrained and auxiliary networks are poorly-calibrated generating highly over-confident predictions while DRL is nearly perfectly calibrated. Overall, considering both label- and distribution-discriminative representations, DRL obtains better performance on all metrics than any of the two components. Therefore, DRL can take advantage of both components, which indicates that label- and distribution-discriminative representations are complementary to each other.

## 4.6 Visualization

We apply Grad-Cam Selvaraju et al. (2020) to produce coarse localization maps that highlight the essential regions for learning label- and distribution-discriminative representations. Figure 8 visually shows the heat maps of different classes of input samples on pretrained and auxiliary networks. We can observe that the two networks focus on different regions for the same input sample. For example, for the malamute, the pretrained network focuses on the head, while the auxiliary network focuses on the body. According to the design principles of the two networks, the pretrained network is trained without any constraint on label-discriminative representations, while the auxiliary network is trained with an implicit constraint to ensure that the learned distribution-discriminative representations differ from the corresponding label-discriminative representations. Therefore, we know that the pretrained network tends to extract strongly label-related information from an ID sample to learn a label-discriminative representation which is the head information in the malamute. Conversely, the auxiliary network tends to extract the weakly label-related information to learn a distribution-discriminative representation which is the body information in the malamute. According to the presented experimental results and the properties of the two networks, the pretrained and auxiliary networks extract strongly and weakly label-related information from inputs to learn representations, respectively. For an ID sample, the label- and distribution-discriminative representations that correspond to the same label are complementary.

# 5 Conclusion and Future Work

In this paper, we propose Dual Representation Learning (DRL) method combining both label- and distribution-discriminative representations to improve the OOD sensitivity. From the modeling perspective, a pretrained network learns label-discriminative representations that are strongly related to labeling, while an auxiliary network learns complementary distribution-discriminative representations that are weakly related to labeling. From the data perspective, the label- and distribution-discriminative representations are complementary in labeling for an ID sample and correspond to different labels for an OOD sample. Based on the different informativeness properties of ID and OOD samples, DRL distinguishes ID and OOD samples according to the OOD scores estimated by integrating the two representations. We empirically demonstrate that DRL more effectively detects OOD samples than the state-of-the-art post-hoc, confidence enhancement and ensemble methods across various datasets. This work makes the first attempt to improve the OOD sensitivity by learning both label and distribution-sensitive representations from training ID samples. However, it pays the price to learn an extra network built on a pretrained network. A riveting follow-up is to explore the OOD-sensitive information by augmenting ID samples and then finetuning a pretrained network with the augmented ID samples.

## Acknowledgments

This work was supported in part by the Australian Research Council Discovery under Grant DP190101079 and in part by the Future Fellowship under Grant FT190100734.

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
