# OpenReview forum: "Dual Representation Learning for Out-of-distribution Detection"
_TMLR — Accepted by TMLR_

### Review · Reviewer_Z9uQ · 2023-05-19

**Summary Of Contributions:**

This paper works on out-of-distribution detection and proposes Dual Representation Learning (DRL) to distinguish in- and out-of-distribution samples. The motivation is to make out-of-distribution samples harder to have high-confidence predictions by exploring both strongly and weakly label-related information from in-distribution samples. For a pretrained network with label-discriminative representations, an auxiliary network is trained to explore the remaining weakly label-related information to learn distribution-discriminative representations. That is, for a label-discriminative representation, its complementary distribution-discriminative representation is constructed by integrating diverse representations. The goal is to combine label- and distribution-discriminative representations to detect out-of-distribution samples. Experiments are conducted on popular benchmark datasets to demonstrate the effectiveness of the proposed DRL for out-of-distribution detection.

**Audience:**

Yes

**Broader Impact Concerns:**

None.

**Claims And Evidence:**

Yes

**Requested Changes:**

Please see the weaknesses above.

**Strengths And Weaknesses:**

Strengths:
1. The motivation is clear, and the idea of DRL is interesting and novel.
2. The reported results are superior to the compared baselines and the ablation studies as well as parameter analysis are sufficient.

Weaknesses:
1. The compared baselines need to be enriched. There are some more recent methods published in 2022 and 2023 that should be introduced and compared, such as "Out-of-Distribution Detection with Deep Nearest Neighbors", "Provable guarantees for understanding out-of-distribution detection" "How to Exploit Hyperspherical Embeddings for Out-of-Distribution Detection?".
2. The theoretical analysis and support is weak. I am interested to see if there is a correlation between the similarity of different datasets and the final performance, such as DTD47 vs. CIFAR10 and CUB200vs. CIFAR10. Further analysis on such results is also insufficient.
3. For visualization, is it possible to show some results on ImageNet? Because CIFAR 10 is relatively simple, the heat maps are easier.

---

> ### Author Response · Authors · 2023-06-21
> **Response to Reviewer Z9uQ**
>
> Thank you for your valuable suggestions and comments, which have been addressed below and in the paper. Please find the point-by-point response below to each of your comments, and kindly let us know if you have any further comments or suggestions.
>
> Q1. Comparison experiments :
>
> Thank you for your suggestion. In this revision, we have discussed the research work you suggested in the related work and evaluated their performance in the experiments. Specifically, compared with GEM [R1], KNN+ [R2] and CIDER [R3], the proposed DRL achieves 2.23%, 3.31% and 2.20% improvement, respectively. This is because the distribution-discriminative representations coupled with the corresponding label-discriminative representations contain more label-related information than any of them, which reduces the prediction confidence for an OOD sample owning minimum label-related information and enhances the prediction confidence for an ID sample owning all the labeling information. It enlarges the confidence gap between ID and OOD samples.
>
> [R1] Morteza et al., Provable guarantees for understanding out-of-distribution detection, AAAI, 2022.
>
> [R2] Sun et al., Out-of-distribution detection with deep nearest neighbors, ICML, 2022.
>
> [R3] Ming et al., How to exploit hyperspherical embeddings for out-of- distribution detection? ICLR, 2023.
>
> Please refer to Section 2 in page 3 and Section 4.2 in page 11 for more information.
>
> Q2. Correlation between the similarity of different datasets:
>
> To address your concern, following your suggestion and the research setup of Winkens et al., we have evaluated the OOD detection performance on near and far samples in this revision. Near and far samples represent the out-of-distribution samples slightly and significantly different from ID samples, respectively. Specifically, for the ID dataset CIFAR10, we adopt CIFAR100 and CUB200 as near OOD datasets because they contain some classes having similar semantics to that of CIFAR10. Furthermore, we adopt Oxfordflowers102 and DTD47 as far OOD datasets because the classes of the two datasets are vastly different from that of CIFAR10. The experimental results are summarized in Table 3. All the considered methods can effectively detect far OOD samples due to the large semantic gap of classes between ID and OOD datasets. Furthermore, DRL achieves 2.62% improvement over the compared methods. Conversely, it is difficult to distinguish near OOD samples from ID samples since the performance of each method is lower than that on far OOD samples. However, DRL still achieves the best detection performance and obtains more significant improvement, i.e., 2.94 %. This is because DRL improves OOD sensitivity by exploring more label-related information from original inputs. Therefore, DRL can leverage more details to describe ID samples, which can more effectively differentiate from OOD samples with different information.
>
> Analyzing OOD detection performance on different datasets theoretically is challenging. This is because it involves how to evaluate the discrepancy between two given datasets, i.e., ID and OOD datasets, which is rarely explored recently. You may want to refer to [R4] and [R5] if you are interested in the advancements in dataset discrepancy measurement which is also our future research direction.
>
> [R4] Liu et al., Comparing distributions by measuring differences that affect decision making, ICML, 2020.
>
> [R5] Zhao et al., Learning deep kernels for non-parametric two-sample tests, ICLR, 2022.
>
> Please refer to Section 4.2.2 in page 12 for more information.
>
> Q3. Visualization:
>
> We totally agree with your opinion. CIFAR10 is a simple data which is unsuitable to highlight the essential regions for learning label- and distribution-discriminative representations. The visualization results reported in this paper are on Mini-ImageNet. We wrongly spelled the name of the dataset in the title of Figure 8 in the previous version, and we have corrected the mistake in this revision. We apologize deeply for the misunderstanding caused by this error on our part. Furthermore, we have made a thorough revision of this paper and a proofreading.
>
> Please refer to Figure 8 in page 16 for more information.

---

### Review · Reviewer_ZE3v · 2023-06-05

**Summary Of Contributions:**

First, this paper argues that pretrained networks learn label-discriminative representations and overlook distribution-discriminative representations, which is a reasonable explanation for their suboptimal performance. Second,  it proposes a novel learning principle of distribution-discriminative representations.

**Audience:**

Yes

**Broader Impact Concerns:**

No ethical concerns.

**Claims And Evidence:**

Yes

**Requested Changes:**

1. On page 8, "Accordingly, the correlation between $f_\theta$ and $z_\theta$ is", "correlation" is a term in statistics. I recommend changing a word to avoid confusion

**Strengths And Weaknesses:**

Strengths:
1. Clear and reasonable motivation for improving OOD performance of pretrained models.
2. The idea of learning complementary representation based on a learned representation sounds interesting.
3. The method seems novel.

Weaknesses
1. I'm convinced that label-discriminative representations are insufficient for OOD tasks since they lose information on original data. But I'm not convinced that the complementary representations should be learned with the same objective ((1) and (6)). I think the author should be more specific to this argument.
2. Equation (14) indicates the use of mean and covariate. But there is no mechanism to ensure z_\theta(x) to be (or learn to be) the mean. So I'm not sure if the deduction of (10)-(13) is necessary and correct.
3. Is label-discriminative representation necessary for OOD tasks since we only need to identify whether a sample is in distribution?

---

> ### Author Response · Authors · 2023-06-21
> **Response to Reviewer ZE3v**
>
> Thank you for your valuable suggestions and comments, which have been addressed below and in the paper. Please find the point-by-point response below to each of your comments, and kindly let us know if you have any further comments or suggestions.
>
> Q1. Objective of complementary representations:
>
> The learning principles for label- and distribution-discriminative representations, i.e., Eq. (1) and Eq. (6), are significantly distinct. To address your concern, we have provided more explanations in this revision. Eq. (1) presents the learning principle for label-discriminative representations which forces a network to explore the strongly label-related information. However, Eq. (6) considers a constraint $\mathcal{I}(\mathcal{D};\mathcal{C})$ which forces a network to explore the label-related information discarded by label-discriminative representations and use them to learn the corresponding distribution-discriminative representations. Therefore, based on the two learning principles, the label-related information of label- and distribution-discriminative representations is different and complementary.
>
> Please refer to the second and fourth paragraphes of Section 3.2 in page 6 for more information.
>
> Q2. Deduction of label-discriminative representations:
>
> To address your concern, we have provided more explanations about these equations in this revision. Overall, an auxiliary network $z_\theta(x)$ is applied to estimate the component representation expectation $\mu_{\mathcal{Z}}$, which is a widely used trick in Bayesian neural networks [R1, R2], and Eqs. (10)-(13) are necessary. We obtain a distribution-discriminative representation by integrating infinite component representations as shown in Eq. (12), where a component representation more similar to the corresponding label-discriminative representation is assigned with a lower weight, or vice versa. This mechanism ensures that the distribution-discriminative representation is significantly different from its corresponding label-discriminative representation. However, solving Eq. (12) is intractable. This is because it is impossible to model infinite component representations. We notice that Eq. (12) depends on the component representation expectation, which indicates that we can apply a single network to model the component representation expectation $\mu_{\mathcal{Z}}$ rather than infinite networks to model the component representations. Based on the derived result in Eq. (12), we apply an auxiliary network $z_\theta(x)$ to approximate the component representation expectation $\mu_{\mathcal{Z}}$ and obtain the Eq. (14). Therefore, Eq. (14) can be treated as an implicit constraint for learning distribution-discriminative representations differing from its corresponding label-discriminative representations.
>
> [R1] Gal et al., Dropout as a bayesian approximation: Representing model uncertainty in deep learning, ICML, 2016.
>
> [R2] Ardywibowo et al., VFDS: variational foresight dynamic selection in bayesian neural networks for efficient human activity recognition, AISTATS, 2022.
>
> Q3. Necessity of label-discriminative representations:
>
> To address your concern, in this revision, we have run a set of ablation study to verify that label-discriminative representations are necessary for the OOD detection task. The results in Figure 6 and Figure 7 show that merely exploiting distribution-discriminative representations achieves slightly better performance in detecting OOD samples than the label-discriminative representations. This is because the distribution-discriminative representation of an ID sample contains information that is weakly related to its labeling, and the weakly label-related information is more sensitive to OOD samples than the strongly-related information in the label-discriminative representations. However, combining both label- and distribution-discriminative representations can achieve the best OOD detection performance. This is because the two represents are complementary, the combination contains more label-related information than any of them. It reduces the prediction confidence for an OOD sample owning minimum label-related information and enhances the prediction confidence for an ID sample owning all the labeling information, which enlarges the confidence gap between ID and OOD samples.
>
> Q4. Avoid confusion:
>
> Thank you for your suggestion. Following your suggestion, we have replaced "correlation" with "relationship" in this revision. Furthermore, we have made a thorough revision of this paper and a proofreading.

---

### Review · Reviewer_5W2A · 2023-06-12

**Summary Of Contributions:**

This paper introduces a novel method, Dual Representation Learning (DRL), for out-of-distribution (OOD) detection. The main idea is to make OOD samples more challenging to be assigned with high-confidence predictions by learning from both strongly and weakly label-related information. DRL trains an auxiliary network to explore weakly label-related information and learn distribution-discriminative representations. The paper demonstrates that DRL outperforms state-of-the-art methods in OOD detection.

**Audience:**

Yes

**Claims And Evidence:**

Yes

**Requested Changes:**

Already discussed with suggestions in the section above.



**Strengths And Weaknesses:**

Strengths
- The paper addresses an important issue in the field of OOD detection, offering an innovative approach that builds upon the exploration of both strongly and weakly label-related information.
- The concept of DRL is unique and appears to be effective based on the experimental results provided.
- The authors have done a commendable job in providing a clear and detailed explanation of the methodology and the corresponding results.
- The research seems to be well-motivated and theoretically grounded, offering an interesting extension to the current body of knowledge in the area of OOD detection.

Weaknesses:
- While the experimental results indicate that DRL outperforms the state-of-the-art methods, a more comprehensive comparison and deeper analysis might strengthen this claim. For example, detailed discussions about why DRL works better and under what conditions it might not perform as well could be valuable.
- There could be a more theoretical discussion about the information bottleneck principle, including its limitations, as it forms a crucial part of the research foundation.
- Do you have conducted a sanity check of the Distribution-discriminative Representation? Specifically, whether the distribution-discriminative representations are still useful for discriminating the labels of input data? Is it complementary to label-discriminative representations in the sense of ordinary classification? Based on the motivation of the paper, the distribution-discriminative representations would be useless in classification.

Overall, this paper represents a valuable contribution to the field of OOD detection, introducing a novel methodology that seems to be effective based on the experimental results. With a few improvements, particularly in terms of clarity and depth of explanation, the paper could be even stronger.

---

> ### Author Response · Authors · 2023-06-21
> **Response to Reviewer 5W2A**
>
> Thank you for your valuable suggestions and comments, which have been addressed below and in the paper. Please find the point-by-point response below to each of your comments, and kindly let us know if you have any further comments or suggestions.
>
> Q1. Comprehensive comparison and deeper analysis :
>
> Following your suggestion, in this revision, we have compared the proposed DRL method with more state-of-the-art methods (GEM, KNN+ and CIDER) to verify its effectiveness, evaluated the performance on near and far OOD samples, and provided deeper analyses for the significant improvement.
>
> The experimental results in Table 1 and Table 2 show that DRL can outperform the state-of-the-art methods. This is because DRL leverages more label-related information by label- and distribution-discriminative representations, which reduces the prediction confidence for an OOD sample owning minimum label-related information and enhances the prediction confidence for an ID sample owning all the labeling information. It enlarges the confidence gap between ID and OOD samples. However, the mechanism of DRL also indicates that it may fail if the training ID samples contain little label-related information with numerous label-unrelated information. This is because training on these ID samples causes no remaining label-related information that the auxiliary network of DRL can explore.
>
> Furthermore, we have evaluated the performance on near and far OOD samples in this revision. Near and far samples represent the out-of-distribution samples slightly and significantly different from ID samples, respectively. The results summarized in Table 3 show that DRL can achieve significant improvement over the state-of-the-art methods on both ID and OOD samples, i.e., 2.62% and 2.94% improvements on far and near OOD datasets, respectively. This is because DRL improves OOD sensitivity by exploring more label-related information from original inputs. Therefore, DRL can leverage more details to describe ID samples, which can more effectively differentiate from OOD samples with different information.
>
> Please refer to Section 4.2.2 in page 12 for more information.
>
> Q2. Discussion about the information bottleneck principle:
>
> Following your suggestion, we have discussed the advantages and limitations of the information bottleneck principle in this revision. The information bottleneck principle offers a principled approach to encourage the learning of more abstract and invariant representations from input data for a given task, which forces networks to capture the underlying structures and dependencies in the data that are relevant to this task. Accordingly, using different constraints, we can obtain the learning principles for label- and distribution-discriminative representations, as shown in Eq. (1) and Eq. (6), respectively. Specifically, label-discriminative representations aim to capture the strongly label-related information, while distribution-discriminative representations aim to capture the remaining weakly label-related information. However, the information bottleneck principle finds a trade-off between compression and loss of information. Therefore, exploring all the label-related information is impossible, and some essential details in the data may be discarded.
>
> Please refer to the last paragraph of Section 3.2 in page 8 for more information.
>
> Q3. Sanity check of distribution-discriminative representations:
>
> To address your concern, in this revision, we have run a set of ablation study to analyze the ID classification and OOD detection performance of label-discriminative representations, distribution-discriminative representations, and their combination. The results in terms of accuracy and AUROC are presented in Figure 6 and Figure 7. They show that the distribution-discriminative representations perform slightly worse in classifying ID samples than the label-discriminative representations. It indicates that the distribution-discriminative representations are still useful in classification, which matches our motivations. This is because a single label-discriminative representation cannot explore all the label-related information according to the learning principle in Eq. (1). Furthermore, the distribution-discriminative representation explores the weakly label-related information which is discarded by the label-discriminative representations according to the learning principle in Eq. (6). Therefore, the label information of the two representations is complementary. It can also explain that combining label- and distribution-discriminative representations can improve ID classification and OOD detection performance. Specifically, considering more label-related information can improve the generalization ability and make OOD samples harder to be assigned with high-confidence predictions.
>
> Please refer to Section 4.5 in page 17 for more information.

---

### Comment · Reviewer_Z9uQ · 2023-07-12
**recommendation**

In the revised version, the authors have addressed my previous concerns. I have no other comments and recommend acceptance.

---

> ### Comment · Action_Editors · 2023-07-18
> **Copy this to correct "Official Recommendation" entry**
>
> Thanks for completing this recommendation.  Openreview has you marked as late on this task because this is posted as a comment instead of an "Official Recommendation."  Could you re-enter it in openreview with the correct form?  This will allow the semi-automated review process to complete properly, and stop us from receiving reminder emails.
>
> Thanks again.

---

### Decision · Action_Editors · 2023-08-16

**Recommendation:** Accept as is

**Comment:**

The reviewers were unanimous that the revised submission meets the criteria for acceptance and addressed open concerns from the initial submission.

**Audience:**

It is clear that the article is within scope.  It is about a core question in machine learning.

**Claims And Evidence:**

The revised version of the submission meets the requirements that evidence supports the claims in the article.  Quoting from Reviewer 5W2A
> Based on the original manuscript, the authors' responses to my comments, and their revision, I am leaning towards accepting this paper. They have addressed my concerns thoroughly, providing more in-depth analyses and comparisons with state-of-the-art methods, discussing the limitations and benefits of the information bottleneck principle, and performing a sanity check for the Distribution-discriminative Representation.

Reviewers unanimously voted "accept" or "leaning accept"